# Domain-Agnostic Crowd Counting via Uncertainty-Guided Style Diversity Augmentation

## ABSTRACT

Domain shift poses a significant barrier to the performance of crowd counting algorithms in unseen domains. While domain adaptation methods address this challenge by utilizing images from the target domain, they become impractical when target domain images acquisition is problematic. Additionally, these methods require extra training time due to the need for fine-tuning on target domain images. To tackle this problem, we propose an Uncertainty-Guided Style Diversity Augmentation (UGSDA) method, enabling the crowd counting models to be trained solely on the source domain and directly generalized to different unseen target domains. It is achieved by generating sufficiently diverse and realistic samples during the training process. Specifically, our UGSDA method incorporates three tailor-designed components: the Global Styling Elements Extraction (GSEE) module, the Local Uncertainty Perturbations (LUP) module, and the Density Distribution Consistency (DDC) loss. The GSEE extracts global style elements from the feature space of the whole source domain. The LUP aims to obtain uncertainty perturbations from the batch-level input to form style distributions beyond the source domain, which used to generate diversified stylized samples together with global style elements. To regulate the extent of perturbations, the DDC loss imposes constraints between the source samples and the stylized samples, ensuring the stylized samples maintain a higher degree of realism and reliability. Comprehensive experiments validate the superiority of our approach, demonstrating its strong generalization capabilities across various datasets and models. Our code will be made publicly available.

## CCS CONCEPTS

• **Computing methodologies** → **Computer vision**; • **Human-centered computing** → *Collaborative and social computing*.

## KEYWORDS

Crowd Counting, Domain Generalization, Uncertainty, Style Augmentation

## 1 INTRODUCTION

Crowd counting is focused on accurately estimating the number of individuals in crowded scenes and has garnered significant attention in recent years. Accurate crowd counting is essential for understanding crowd behavior, ensuring public safety, and facilitating urban

Permission to make digital or hard copies of all or part of this work for personal or classroom use is granted without fee provided that copies are not made or distributed for profit or commercial advantage and that copies bear this notice and the full citation on the first page. Copyrights for components of this work owned by others than the author(s) must be honored. Abstracting with credit is permitted. To copy otherwise, or republish, to post on servers or to redistribute to lists, requires prior specific permission and/or a fee. Request permissions from permissions@acm.org.

*ACM MM, 2024, Melbourne, Australia*

© 2024 Copyright held by the owner/author(s). Publication rights licensed to ACM.
ACM ISBN 978-x-xxxx-xxxx-x/YY/MM
https://doi.org/10.1145/nnnnnnn.nnnnnnn

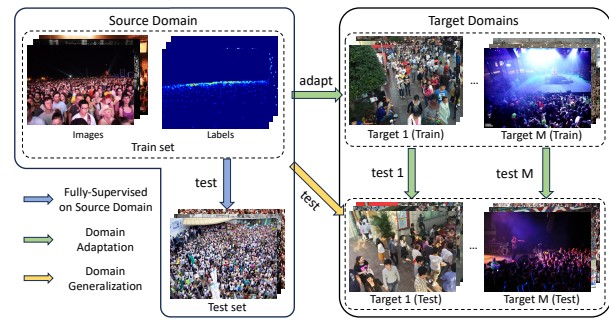

**Figure 1: Comparison between fully-supervised on source domain, domain adaptation, and domain generalization. For better visual effects, the images are cropped and scaled.**

planning across various domains. In the past decade, with the rapid advancement of artificial intelligence, crowd counting has also made remarkable progress [21, 31, 33, 35, 39, 44, 62, 65]. Almost all of these methods assume that the training and test sets are drawn i.i.d. from the same underlying distribution. However, this assumption is often violated in real-world scenarios due to substantial variations in data acquisition devices, data collection environments, and other factors. Such domain shifts between the source and target domains can lead to significant performance degradation.

To alleviate this issue, several studies attempt to incorporate Domain Adaptation (DA) techniques within the realm of crowd counting [7, 16, 37, 50, 52, 53, 69]. These methods aim to minimize the domain gap by learning domain-invariant feature representations or style information specific to the target domain. While these methods yield promising results, their efficacy depends on access to images in the target domain. However, obtaining sufficient data from target domains may encounter numerous challenges, especially as people become increasingly aware of privacy protection. What is more, it is troublesome and time-consuming to finetune the pre-trained model for each target domain.

As shown in Fig. 1, compared to fully-supervised on source domain methods that perform worse on target domain, and DA methods that necessitate target domain images, Domain Generalization (DG) techniques can achieve commendable performance without any access to the target domain. Meta-learning and data augmentation are two important categories in the domain generalization. MLDG [26] proposes to use meta-train and meta-test sets to simulate the domain shifts to learn domain-invariant features. Building on MLDG, DGCC [8] employs meta-learning technique for domain generalization in crowd counting. However, DGCC suffers from a complex network structure and sensitive hyperparameters, and it requires maintaining an additional domain-invariant crowd memory which adversely impacts inference speed. AdaIN [19] is a data augmentation technique that can transmute images into any arbitrary style while preserving content. This capability enables the network to

learn from a more diversified set of data within the source domain and consequently achieve a more general model. Leveraging AdaIN, we first propose to use data augmentation technique in crowd counting, which can bolster the generalization ability of models without impacting the network architecture and inference speed.

In this paper, we propose an Uncertainty-Guided Style Diversity Augmentation (UGSDA) method for domain-agnostic crowd counting. The proposed UGSDA method comprises a Global Style Element Extraction (GSEE) module, a Local Uncertainty Perturbation (LUP) module and the Density Distribution Consistency (DDC) loss. Inspired by [63], the GSEE module represents the $C$-dimensional style space by sampling $C$ global style elements. To address the potential limitations of source domain data in capturing all possible styles, we employ the LUP during the generation process of each training batch to introduce uncertainty into the global style elements, thereby enabling the creation of a diverse set of styles. LUP calculates the mean and variance at the batch level for the images, and samples from a Gaussian distribution to obtain perturbation values for the global style elements. To prevent the stylized samples from deviating too much from real-world scenarios, we also employ the proposed DDC loss to regularize the difference between the density distribution of real samples and stylized samples in the high-dimensional space. These components together generate sufficiently diverse samples during the training process, thereby enhancing the model's generalization capabilities. The contributions of this paper are summarized as follows:

- To our best knowledge, this is the first attempt to employ data augmentation techniques for domain generalization in the field of crowd counting, offering advantages in terms of simply and plug-and-play functionality without affecting inference speed.
- The proposed uncertainty-guided style diversity augmentation method dynamically generates diverse training samples by combining global style elements extracted from the entire source domain with slight perturbations obtained from each training batch. Additionally, the density distribution consistency loss effectively optimizes the realism of the density distribution of the stylized samples.
- Comprehensive experiments on different data sets validate that our method can achieve superior performance to the state-of-the-art algorithms. Moreover, our method exhibits strong generalization capabilities across various crowd counting networks.

## 2 RELATED WORKS

### 2.1 Crowd Counting

**Fully-Supervised Crowd Counting.** After the initial proposition of density learning methods by Lempitsky et al. [25], the crowd counting research has pivoted from detection-based [13] and regression-based [2, 48] paradigms to methods reliant on density maps. Owing to the physical law of foreshortening, which dictates that objects appear smaller as they recede into the distance, the generated Gaussian density maps may confront significant scale variations. One type of approach commonly adopted to address this issue involves refining network design for more accurate density map estimation. Specifically, some works [4, 45, 57, 62] employ multi-column network

architectures to learn feature information across diverse scales. Some studies endeavor to utilize scale-selection [9, 15, 43, 46] techniques or learn to scale [54, 55] methodologies to mitigate the challenges posed by scale variations. Some methods leverage attention mechanisms [1, 30, 32, 41] or deformable convolution [14, 64] to enhance feature representational capacity. Another type of approach elevates the accuracy of crowd counting by innovating how to generate better ground truth density maps. Specifically, Zhang et al. [60] generate density maps by employing perspective normalization to generate perspective maps, which are then amalgamated with the center positions of pedestrian heads. MCNN [62] introduced geometry-adaptive kernels as a replacement for a fixed Gaussian kernel. Wang et al. [50] try to employ attention mechanisms for the adaptive fusion of density maps generated from distinct, predefined Gaussian kernels. Huang et al. [18] and ADMAL [7] propose a method for adaptively generating density maps based on the spatial features of the objects. However, most of these works neglect the domain shift that often occurs between training and testing scenarios in many real-world applications, leading to a significant degradation in model performance.

**Domain Adaptive Crowd Counting.** To counteract the potential domain shifts, Domain Adaptation (DA) techniques have been introduced into crowd counting [7, 36, 50, 67, 68]. Within the DA setting, the network requires target domain images for fine-tuning. [7, 12, 16, 71] all employ adversarial learning at the feature level to force the network to learn domain-invariant features. There are other approaches [10, 50, 51] that deploy adversarial learning techniques at the image level. The crux of this type of method lies in employing the Generative Adversarial Network (GAN) [70] principle to transfer the stylistic attributes of source domain images to the target domain, thereby diminishing the domain shift between the source and target domain. Apart from adversarial learning approaches, Liu et al. [36] use uncertainty estimation to facilitate self-supervised learning in target domain, thereby accomplishing fine-tuning purposes. DAOT [69] proposes a domain-agnostically aligned optimal transport strategy that aligns domain-agnostic factors and managing outliers.

### 2.2 Domain Generalization

Different from domain adaptation methods, which can access both source domain and target domain images, domain generalization methods are limited to obtaining data solely from the source domain. A straightforward approach is to enhance the diversity of source domain data through data augmentation [17, 27, 56, 58]. SHADE [63] proposes a style hallucination module to generate new style-diversified data to improve generalization ability and relieve the model from overfitting in training domains. DSU [28] proposes to use the feature statistics of the training data to improve the generalization ability of model. Another prevalent approach is to leverage the meta-learning technique [8, 23, 26] to align the distribution of the source domain by learning domain-invariant representations. The essence of this category of work is to divide the source domain into meta-train and meta-test sets, thereby simulating the domain shift between the source and target domains.

Building on MLDG [26], DGCC [8] employs meta-learning technique in conjunction with domain-invariant and -specific crowd memory modules to achieve domain generalization in crowd counting. Unlike the DGCC method, we propose an uncertainty-guided

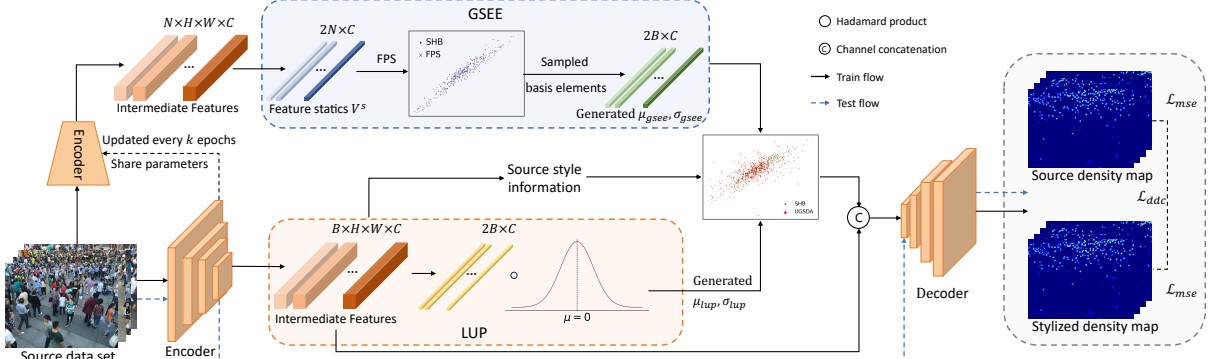

**Figure 2: Architecture of our uncertainty-guided style diversity augmentation method. The sampling results within Shanghai Tech B (SHB) data set have been visualized to underscore the efficacy of our proposed UGSDA method. Features represented by the same color represent that they are obtained by repeating. Details of the GSEE and LUF can be referred to Section 3.**

style diversity augmentation method that generates a diverse set of samples to prevent the network from overfitting to the source domain. In addition, compared to DGCC, our method introduces no modifications to the network architecture and has no impact on inference time. To our best knowledge, this is the first attempt to address domain generalization issues in crowd counting through data augmentation techniques.

## 3 METHODOLOGY

### 3.1 Framework and Preliminary Concepts

**Setting and Overview.** Assuming that we have an observable source domain $\mathcal{D}^s = \{(x_i^s, y_i^s)\}_{i=1}^N$, where $N$ is the number of images in domain $\mathcal{D}^s$ and $(x_i^s, y_i^s)$ denotes the image-label pair for the $i$-th sample in source domain. Besides, we have $M$ unseen target domains $\{D_i^t\}_{i=1}^M$. Our objective is for the model $\theta$, which is well-trained on the source domain $\mathcal{D}^s$, to also perform well concurrently across these $M$ target domains without any finetuning. Moreover, the proposed method is desired to be easily incorporated into crowd counting algorithms without impacting inference speed.

To achieve our objective, an Uncertainty-Guided Style Diversity Augmentation (UGSDA) method is proposed, which consists of a Global Styling Elements Extraction (GSEE) module and a Local Uncertainty Perturbations (LUP) module. Thanks to these collaborative modules, the proposed UGSDA method can generate sufficiently stylized samples, thereby enhancing the domain generalization capabilities of crowd counting methods. Furthermore, a Density Distribution Consistency (DDC) loss is proposed to constrain the realism of the density distribution of the stylized samples in the high-dimensional feature space. The overall framework is shown in Fig. 2.

**Preliminaries.** A model trained on one domain may struggle to adapt to new domains. Data augmentation is a strategy that can expand the amount and diversity of data. Since the network is exposed to data in various styles, it can understand invariances across more data domains [6]. For instance, image flipping enables the network to deal with perspective effects in crowd counting [36]. That is why use style variation methods can help improve the network's generalization ability. However, some image data augmentation methods, such as

Rotation, Photometric Transformer, and Mixup [61], are essentially predefined manipulations or combinations of operations applied to images. These data augmentation techniques fall short in augmenting the variety of data styles and distributions, such as acquiring a new style or altering the distribution ratio of styles within the dataset.

In some generative models [19, 22], it has been found that the use of Adaptive Instance Normalization (AdaIN) [19] allows for adaptation to arbitrarily given styles. The formulation for AdaIN can be expressed as follows:

$$AdaIN(x, y) = \sigma(y)(\frac{x - \mu(x)}{\sigma(x)}) + \mu(y) \quad (1)$$

where $x$ is content input, $y$ is style input, $\mu(*)$ and $\sigma(*)$ represent the mean and standard deviation across spatial locations, respectively. Through AdaIN, we can scale the normalized content of input using $\sigma(y)$ and shift it using $\mu(y)$, thereby obtaining a new sample that conforms to the style of $y$.

In domain generalization of crowd counting, we have the content $x$ from source domain. To achieve our objectives, the key challenge is to generate real and diverse stylized samples to enhance the model's generalization capability. In Section 3.2, we will discuss how to generate global style elements within the scope of the source domain and Section 3.3 introduces how to generate more out-of-distribution stylized samples with the help of the proposed LUP module.

### 3.2 Global Style Element Extraction

**Method.** As shown in Fig. 1, it can be observed that across various crowd counting data sets, there may exist different style transformations, such as night, daytime, various weather conditions, indoor lighting, etc. It is challenging to manually define all possible style information that may be contained in every unseen target domain. Following [63], we leverage the definition of basis in linear algebra to capture the global style elements within the source domain. To handle out-of-distribution styles, we further employ the Local Uncertainty Perturbation (LUP) module, detailed in Section 3.3.

A basis $S$ is a set of linearly independent vectors in a vector space $V$, through which every vector in that space can be represented. According to this definition, assuming that the style space is a subspace

of a $C$-dimensional vector space, it is feasible to represent the entirety of styles using $C$ linearly independent basis vectors. However, directly using orthonormal vectors ignores the real styles contained in the source domain, which may yield an excess of spurious styles, adversely affecting the model's performance. It should be noted that with source domain data serving as prior knowledge, sampling basis vectors $S^s$ from the style space of the source domain is likely to yield more real new styles. Thanks to the research in style transfer [19, 22], we can represent the style space $V^s$ using the mean and variance of the source domain image features $f^s = \varphi(x^s)$ as follows:

$$V^s = \{(\mu_c(f_i^s), \sigma_c(f_i^s)) \mid 1 \le i \le N, 1 \le c \le C\} \quad (2)$$

where $\mu_c(*)$ and $\sigma_c(*)$ represent the mean and standard deviation across channel dimension, $f_i^s$ denotes the feature map obtained from processing the $i$-th image by the encoder $\varphi(*)$. Within the defined style vector space $V^s$, we can utilize a sampling method to obtain $C$ basis styles every $k$ epochs. We sample the combination weight $W = [w_1, \ldots, w_C]$ from Dirichlet distribution $B([\alpha_1, \ldots, \alpha_C])$ with the concentration parameters $[\alpha_1, \ldots, \alpha_C]$ all set to $1/C$. Then the generated styles $\mu_{gsee}$ and $\sigma_{gsee}$ can be expressed as follows:

$$\mu_{gsee}(x^s) = W \cdot \mu_{basis}(x^s) = W \cdot Sampling(\mu_c(f^s)) \quad (3)$$

$$\sigma_{gsee}(x^s) = W \cdot \sigma_{basis}(x^s) = W \cdot Sampling(\sigma_c(f^s)) \quad (4)$$

where $\mu_{basis}$ and $\sigma_{basis}$ are the $C$ global style elements (basis styles).
**Discussion.** For the sampling method in GSEE, our objective is to obtain global style elements as basis $S$ from the style space $V^s$ of the source domain. Thus, we can employ various sampling methods, such as K-means and Farthest Point Sampling (FPS) [42]. In Section 4.4, we compare and analyze the efficacy of various sampling methods through quantitative experiments and select the FPS method as the optimal approach.

For sampling weights method, we can see from Equations 3 and 4, the sampling weights are utilized to obtain $\mu_{gsee}$ and $\sigma_{gsee}$ through the global style elements. So, it is imperative to ensure that the sampling weights are non-negative and normalized. Non-negativity avoids invalid real style inversion, while normalization balances the contribution of each element, ensuring numerical stability representing style statistical characteristics. A commonly employed distribution that satisfies these two properties is the Dirichlet distribution. In Section 4.4, we validate the rationality and effectiveness of employing this distribution through quantitative experiments.

## 3.3 Local Uncertainty Perturbation

**Method.** Utilizing only the $\mu_{gsee}$ and $\sigma_{gsee}$ obtained from the GSEE may fail to address out-of-distribution scenarios effectively. This is attributable to two reasons: i) styles generated by the GSEE module are likely to be more densely concentrated near the source domain styles; ii) the target domains are entirely unseen, and the source domain's style space is just a small subspace within the totality of style spaces. Fig. 4 validates the existence of these two issues.

A LUP module is thus proposed to complement GSEE. This LUP module aims to represent potential domain shifts by introducing randomness to generate new mean and variance statistics for feature representation. In the LUP, we posit that the distribution of the mean and variance feature statistics adheres to a standard Gaussian distribution. Then, we can model the uncertainty of domain differences

through a standard Gaussian distribution $\mu_{un}, \sigma_{un} \sim \mathcal{N}(0, 1)$. However, due to the disparate scales of the mean and standard deviation of feature samples obtained in each batch, directly employing the sampled uncertainties $\mu_{un}$ and $\sigma_{un}$ may result in the network generating samples that are not realistic. Similarly to [28], in order to scale the sampled uncertainties to the same level as the real feature statistics in the source domain's features, we normalize $\mu_{un}$ and $\sigma_{un}$ using the standard deviation among the source domain feature samples within the same batch. So, the generated styles uncertainties $\mu_{lup}$ and $\sigma_{lup}$ can be expressed as follows:

$$\mu_{lup}(x^s) = \mu_{un} \circ \sigma_b(\mu(f^s)) \quad (5)$$

$$\sigma_{lup}(x^s) = \sigma_{un} \circ \sigma_b(\sigma(f^s)) \quad (6)$$

where $\sigma_b(*)$ represents the standard deviation across batch dimension, $\circ$ denotes the Hadamard product.
**Constraint.** As discussed in the part of method, due to the invisibility of the target domain, we simulate out-of-distribution scenarios by normalizing a standard Gaussian distribution with information from the source domain. However, this strong assumption of a Gaussian distribution under weak constraints (normalization) may result in the network producing an excessive number of unrealistic style images, thereby negatively impacting network performance. Therefore, we additionally propose a Density Distribution Consistency (DDC) loss $\mathcal{L}_{ddc}$ to constrain this issue.

Given the need to consider both the diversity and realism of data styles, we do not directly impose our DDC loss function on the discrepancy between generated and source real styles. Instead, we constrain the outcomes obtained from different stylized images. This is because if a style falls outside the distribution range, the resulting error could be significant. We utilize the Jensen-Shannon Divergence as a loss function to assess the posterior probability between the generated styles and the source styles results as follows:

$$\mathcal{L}_{ddc}(x^s, \hat{x}^s) = \frac{1}{2}(KL(p(\psi(\varphi(x^s)))||A) + KL(p(\psi(\hat{x}^s))||A)) \quad (7)$$

where KL means Kullback-Leibler Divergence, $A = \frac{1}{2}(p(\psi(\varphi(x^s))) + p(\psi(\hat{x}^s)))$, $\hat{x}^s$ means the generated new stylized samples and $\psi$ is the decoder.

## 3.4 Implementation

**Data Augmentation.** We observe that in crowd counting, the stylistic disparities between domains often manifest in the form of weather and illumination variations across different scenes. Compared to the domain shift between real-world and synthetic scenes, these shifts are relatively minor. To ensure the realism of the styles generated by the UGSDA method, we blend the feature statistics representative of the real data's styles from each batch with the newly generated styles. In Section 4.4, we further demonstrate the efficacy of the blending operation. To ensure computational simplicity, we obtain the final style by simply averaging as follows:

$$\mu(x^s) = mean(\mu_{gsee}(x^s) + \mu_{lup}(x^s), \mu(x^s)) \quad (8)$$

$$\sigma(x^s) = mean(\sigma_{gsee}(x^s) + \sigma_{lup}(x^s), \sigma(x^s)) \quad (9)$$

By replacing the style representation $\mu(y)$ and $\sigma(y)$ in Eq. 1 with newly generated styles $\mu(x^s)$ and $\sigma(x^s)$, we can generate stylized samples $\hat{f}^s$ aligned with the content of the input from the source domain.

**Table 1: Quantitative comparisons of domain generalization with SHA and SHB as the source domains. SI&L means source domain images and labels and TI means the target domain images. Underline indicates the best result among algorithms that require TI, while bold denotes the best result among algorithms that do not require TI. Small MAE and small MSE indicate good performance.**

| Method | SI&L | TI | Paradigm | SHA | | | | SHB | | | |
|---|---|---|---|---|---|---|---|---|---|---|---|
| | | | | SHB | | UCF_QNRF | | SHA | | UCF_QNRF | |
| | | | | MAE | MSE | MAE | MSE | MAE | MSE | MAE | MSE |
| SE CycleGAN [50] | ✓ | ✓ | DA | 19.9 | 28.3 | 230.4 | 384.5 | 123.4 | 193.4 | 230.4 | 384.5 |
| SE+FD [16] | ✓ | ✓ | DA | 16.9 | 24.7 | 221.2 | 390.2 | 129.3 | 187.6 | 221.2 | 390.2 |
| RBT [37] | ✓ | ✓ | DA | 13.4 | 29.3 | 175.0 | 294.8 | 112.2 | 218.2 | 211.3 | 381.9 |
| C$^2$MoT [52] | ✓ | ✓ | DA | 12.4 | 21.1 | 125.7 | 218.3 | 120.7 | 192.0 | 198.9 | 368.0 |
| SaKnD [53] | ✓ | ✓ | DA | 17.1 | 27.7 | 120.2 | 217.7 | 137.2 | 224.2 | 184.5 | 320.5 |
| FSIM [68] | ✓ | ✓ | DA | 11.1 | 19.3 | 105.3 | 191.1 | 120.3 | 202.6 | 194.9 | 324.5 |
| MCNN [62] | ✓ | ✗ | NoAdapt | 85.2 | 142.3 | – | – | 221.4 | 357.8 | – | – |
| DSSINet [34] | ✓ | ✗ | NoAdapt | 21.7 | 37.6 | 198.7 | 329.4 | 148.9 | 273.9 | 267.3 | 477.6 |
| BL [38] | ✓ | ✗ | NoAdapt | 15.9 | 25.8 | 166.7 | 287.6 | 138.1 | 228.1 | 226.4 | 411.0 |
| DMCount [47] | ✓ | ✗ | NoAdapt | 23.1 | 34.9 | 134.4 | 252.1 | 143.9 | 239.6 | 203.0 | 386.1 |
| D2CNet [3] | ✓ | ✗ | NoAdapt | 21.6 | 34.6 | 126.8 | 245.5 | 164.5 | 286.4 | 267.5 | 486.0 |
| SASNet [46] | ✓ | ✗ | NoAdapt | 21.3 | 33.2 | 211.2 | 418.6 | 132.4 | 225.6 | 273.5 | 481.3 |
| MAN [32] | ✓ | ✗ | NoAdapt | 22.1 | 32.8 | 138.8 | 266.3 | 133.6 | 255.6 | 209.4 | 378.8 |
| DG-MAN [40] | ✓ | ✗ | DG | 17.3 | 28.7 | 129.1 | 238.2 | 130.7 | 225.1 | 182.4 | 325.8 |
| DGCC [8] | ✓ | ✗ | DG | 12.6 | 24.6 | 119.4 | 216.6 | 121.8 | 203.1 | 179.1 | 316.2 |
| Ours | ✓ | ✗ | DG | **11.6** | **24.5** | **117.0** | **194.1** | **113.4** | **180.8** | **178.1** | **306.7** |

**Loss Function.** The loss function for the training of the proposed method consists of two portions. One is the Mean Square Error loss $\mathcal{L}_{mse}$, which is widely used to evaluate the difference between the estimated density maps and the ground truth density maps. The formula is as follows:

$$\mathcal{L}_{mse} = \frac{1}{2B} \sum_{i=1}^{2B} |d_i - Gauss(g_i)|^2 \quad (10)$$

where $d_i$ is the predicted density map, Gauss($\cdot$) is the Gaussian operation and $g_i$ is the annotated dot map. This becomes $2B$ because data volume is doubled each training iteration through data augmentation.

Another is the density distribution consistency loss $\mathcal{L}_{ddc}$ which is introduced in Section 3.3. Finally, the total loss function of our proposed method is defined as:

$$\mathcal{L} = \mathcal{L}_{mse} + \lambda \mathcal{L}_{ddc} \quad (11)$$

where $\lambda$ is a hyperparameter, and we set to 0.1.

**Network Pipline.** During the training phase, input images are initially processed through an encoder network for feature extraction, yielding source domain image features $f^s$. These features, are subsequently processed by both the GSEE and LUF modules, producing two sets of means and variances. Following this, the final styles are sampled utilizing the data augmentation methods depicted in Equations 8 and 9. Finally, stylized samples $\hat{f}^s$ are obtained using Equation 1. Both the $f^s$ and $\hat{f}^s$ are fed into the decoder and the entire network is trained under the constraint of $\mathcal{L}$.

During the testing phase, all target domain images are directly processed through the encoder and decoder components to obtain the final results. The proposed UGSDA method does not participate in the testing phase, thereby not affecting the inference speed.

## 4 EXPERIMENTS

### 4.1 Datasets

To evaluate the effectiveness of our method, we conduct domain generalization experiments on four representative datasets including Shanghai Tech A/B [62], UCF_QNRF [20] and NWPU [49].

**Shanghai Tech A/B.** The ShanghaiTech dataset comprises two subsets, ShanghaiTech Part A (SHA) and B (SHB). SHA has a total of 482 images of varying resolutions; each image has 501 individuals on average, making it a notably congested dataset. The resolution of SHB is $768 \times 1024$, and each image contains only 123 individuals on average, making it comparatively sparser than the SHA dataset.

**UCF_QNRF.** This dataset contains 1,535 images with a total of 1,251,642 head annotations. In this dataset, 1201 images are the training set, and the remaining 334 images are the test set.

**NWPU.** NWPU is a large-scale crowd counting dataset comprising 5,109 images annotated with 2.13 million labeled points. In this dataset, 3,109 images are split for training, 500 images comprise the validation set, and the remaining 1,500 images, with hidden annotations, serve as the test set. We obtain the performance by submitting the counting results on the test set to the NWPU evaluation system.

### 4.2 Experimental setting

**Implementation Details.** The same backbone as DGCC [8] is used to compare our method with the earlier algorithm fairly. For density estimation, we employ two convolutions to obtain feature maps with the same number of channels as DGCC, and then use a structure identical to DGCC's to predict the density map. The Adam [24] algorithm is used to optimize the network. Hyperparameters $\lambda$ and $k$ are set to 0.1 and 3. All experiments are conducted on Nvidia 3090

**Table 2: Quantitative comparisons of domain generalization with SHA, SHB, and UCF_QNRF as the source domains. All compared results are copied from [8].**

| Method | Paradigm | SHA | | SHB | | UCF_QNRF | | | |
| | | NWPU [1] | | | | SHA | | SHB | |
| | | MAE | MSE | MAE | MSE | MAE | MSE | MAE | MSE |
|---|---|---|---|---|---|---|---|---|---|
| DMCount [47] | NoAdapt | 146.9 | 563.8 | 191.6 | 747.4 | 73.4 | 135.1 | 14.3 | 27.5 |
| SASNet [46] | NoAdapt | 158.8 | 588.0 | 195.7 | 716.8 | 73.9 | 116.4 | 13.0 | 22.1 |
| MAN [32] | NoAdapt | 148.2 | 586.5 | 193.6 | 802.5 | 67.1 | 122.1 | 12.5 | 22.2 |
| DGCC [8] | DG | **143.1** | 567.6 | 175.0 | 688.6 | 67.4 | 112.8 | 12.1 | 20.9 |
| Ours | DG | 150.7 | **535.0** | **159.4** | **571.0** | **65.8** | **104.0** | **10.9** | **19.1** |

GPUs and the batch size is set to 4. The ground truth density maps for SHA, SHB and UCF_QNRF are generated using code from C-3 framework [11] within MATLAB. Following the previous work [49], we produce corresponding ground truth density maps for NWPU.
**Evaluation Metrics.** To quantitatively assess the performance of each model, we adopt two widely used metrics in crowd counting: Mean Absolute Error (MAE) and Mean Squared Error (MSE).

## 4.3 Comparison with State-of-the-art Methods

**Quantitative comparison.** In Table 1, we conduct experiments with SHA and SHB as the source domains, and SHB and UCF_QNRF or SHA and UCF_QNRF as the target domains. We categorize the experimental results into three paradigms: Domain Adaptation (DA), Fully-supervised on source domain(NoAdapt), and Domain Generalization (DG), with only DA undergoing finetuning on the target domain images. Compared to the DA methods requiring target domain images in the first part of Table 1, our approach achieves comparable performance. Notably, when SHB serves as the source domain, our method reaches the best performance without the need for finetuning on target domains.

As shown in the second part of Table 1, compared to fully supervised on source domain crowd counting methods, our method achieves the best results across all four domain generalization settings. These results demonstrate that while crowd counting methods may perform well in the source domain, they are not universally applicable across different domains.

Compare with DG-MAN, which is a domain generalization result obtained from DGCC through the integration of a crowd counting method MAN [32] and a domain generalization method Agr-Sum [40]. It is apparent that although their performance improved in new target domains, there remains a significant gap compared to our method. This suggests that simply incorporating domain generalization techniques into crowd counting methods is inadequate; tailored design to the crowd counting methods is necessary.

Compared with a tailor-designed DGCC, which uses meta-learning technique for crowd counting domain generalization, our method achieved the best results in all four target domains. A potential reason is that their method does not essentially increase data diversity or assign new tasks to the network. Instead, it splits the source domain into meta-training and meta-test subsets to bolster the network's ability to extract domain-invariant features across domains. In contrast, our method enhances the network's generalization ability to

**Table 3: Quantitative evaluation with different networks.**

| Network | Paradigm | SHB | | UCF_QNRF | |
| | | MAE | MSE | MAE | MSE |
|---|---|---|---|---|---|
| CSRNet [29] | NoAdapt | 26.6 | 38.5 | 171.8 | 313.3 |
| Ours | DG | 13.6 | 25.3 | 135.9 | 224.3 |
| SFCN [50] | NoAdapt | 25.2 | 31.9 | 174.3 | 294.0 |
| Ours | DG | 13.1 | 25.9 | 127.2 | 219.5 |
| SASNet[2] [46] | NoAdapt | 24.3 | 35.7 | 166.0 | 304.8 |
| Ours | DG | **11.6** | **24.5** | **117.0** | **194.1** |

unknown domains by employing data augmentation techniques to expose the network to a broader range of new style images. Furthermore, our approach does not affect the network's inference speed, while DGCC requires an additional DICM module during network inference, which slows down the inference time.

Following DGCC, we also conduct experiments with the same model parameters as detailed in Table 2. All of these results further validate the robust generalization capability of our method in the unseen target domains.

**Qualitative comparison.** Fig. 3 presents a qualitative comparison between DGCC and our method when generalized to the UCF_QNRF dataset. Even in styles that are almost absent from the training set, such as grayscale images, dimly lit scenes, and colorful lighting, our method can predict a high-quality density map that accurately reflects the crowd density and the total number of people in the image. In DGCC, because the network is limited to learning generalizability solely from existing data, their performance suffers in these scenarios, particularly in distant and densely crowded regions.

## 4.4 Ablation Studies

In this section, we validate the robustness of our method across various models, while also discussing the effectiveness of each component and the principles behind the selection of hyperparameters. Furthermore, we analyze the choice of methods such as FPS [42] and KL divergence, validating their rationality and effectiveness.

---

[1]The results on NWPU are obtained through submitting to https://www.crowdbenchmark.com/
[2]This network is modified as describe in Section 4.2.

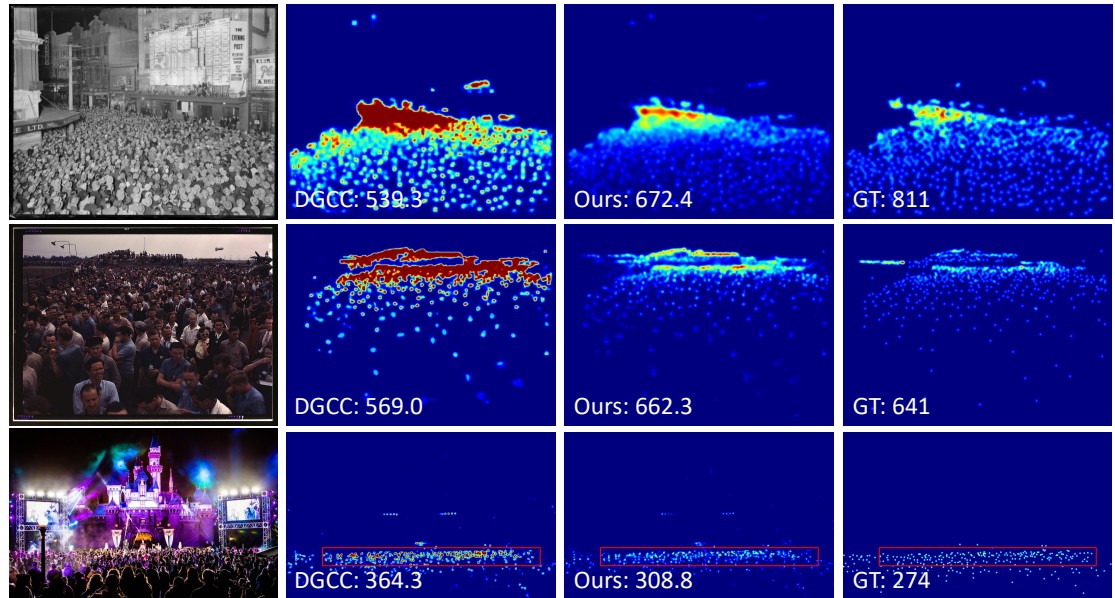

**Figure 3: Qualitative comparison of DGCC and ours. The first column is the original images, while the second, third, and fourth columns show the density maps predicted by DGCC, Ours, and the ground truth, respectively.**

**Robustness to different networks.** In order to prove the effectiveness of our proposed method, we apply it to two other crowd counting networks, CSRNet [29] and SFCN [50]. Taking SFCN as an example, when the model trained on the SHA dataset is directly tested on UCF_QNRF, the MAE and MSE metrics are 174.3 and 294.0, respectively. After incorporating our proposed UGSDA method, the model shows a significant improvement of 47.1 (27.02%) and 74.5 (25.34%) in the MAE and MSE metrics, without finetuning. As shown in Table 3, when employing our method into these networks, there is a varying degree of performance improvement in terms of MAE and MSE. These results prove that our method not only possesses domain generalization capabilities but also exhibits universality across different networks.

**Ablation of GSEE.** We first investigate the impact of the insertion position of GSEE on network performance. GSEE_$i$ denotes the insertion of GSEE after the $i$-th max-pooling layer, where GSEE_1 is the strategy in Ours. As shown in Table 4, the network performance declines when we apply the GSEE module at deeper layers of the network's output. This is because in CNNs, shallow layers typically capture more style-related information, while deeper layers yield more semantic information [19, 59]. Since GSEE needs to capture the global style elements from the source domain features, the performance is optimal in our setup using GSEE_1.

We also give some ablation studies about the sampling method. As shown in Table 4, using k-means for sampling global style elements results in a certain degree of performance degradation than Ours (FPS). This is because the k-means algorithm's insufficient focus on margin styles in the source domain, leading to a more significant performance drop on the UCF_QNRF dataset, which has a larger domain gap with the source domain. For the combination weights, we give the results compare to Gaussian Distribution (GD) which is a common used distribution. As we discussed in Section 3.2,

**Table 4: Ablation studies of GSEE.**

| Method | SHB | | UCF_QNRF | |
|---|---|---|---|---|
| | MAE | MSE | MAE | MSE |
| GSEE_2 | 13.8 | 25.3 | 125.1 | 218.7 |
| GSEE_3 | 14.2 | 26.4 | 140.0 | 245.5 |
| GSEE_4 | 14.7 | 29.1 | 142.4 | 247.8 |
| K-means | 11.9 | 24.9 | 125.8 | 210.3 |
| GD | 15.7 | 29.4 | 146.0 | 247.3 |
| Ours | **11.6** | **24.5** | **117.0** | **194.1** |

GD yielded comparatively poor results because the lack of correct physical interpretation.

**Ablation of LUP.** To examine the effectiveness of the LUP in the proposed method, the network is re-trained with GSEE only named +GSEE. Building upon +GSEE, we progressively adding LUP, and its associated components which proposed to constrain LUP, to validate the efficacy of the introduced of LUP.

As shown in Table 5, a performance decline was observed when only LUP was added, compared to +GSEE. However, this result is not surprising. As analyzed in Section 3.3, this is attributed to the introduction of a hypothetical perturbation by LUP under unconstrained conditions, which is likely to negatively impact performance. Therefore, to impose constraints and balance the diversity and realism of generated styles, we further proposed the DDC loss and the Blend method. It is observable that performance improved upon the individual addition of DDC loss and the Blend method, and the more potent constraining ability of Blend made our method superior to +GSEE. When both of them are incorporated, the model achieves optimal performance. To further validate the effectiveness of the KL

**Table 5: Ablation studies of UGSDA and its components.**

| Method | SHB | | UCF_QNRF | |
|---|---|---|---|---|
| | MAE | MSE | MAE | MSE |
| baseline | 24.3 | 35.7 | 166.0 | 304.8 |
| +GSEE | 12.5 | **22.1** | 125.8 | 213.5 |
| +GSEE, LUP | 14.3 | 28.0 | 145.7 | 267.9 |
| +GSEE, LUP, DDC | 13.4 | 25.2 | 128.0 | 213.8 |
| +GSEE, LUP, Blend | 12.1 | 24.7 | 120.9 | 200.3 |
| LUP $w/o$ Norm | 12.1 | 25.2 | 126.4 | 207.5 |
| DDC $w$ CS | 12.9 | 24.5 | 120.8 | 205.8 |
| All (Ours) | **11.6** | 24.5 | **117.0** | **194.1** |

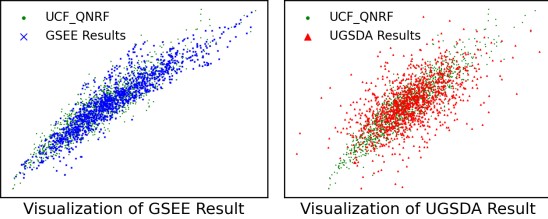

Figure 4: Visualization of GSEE and UGSDA results.

divergence within the DDC loss in rendering our stylized images more realistic by measuring distribution differences, we employ Cosine Similarity (CS) as a substitute to measure the distribution differences. As demonstrated in Table 5 under DDC $w$ CS, employing KL divergence facilitates a more accurate measurement of distribution differences, thereby yielding enhanced performance.

As demonstrated by LUP $w/o$ Norm in Table 5, the absence of scale constraints leads to a decline in performance, underscoring the significance of normalization. Intuitively, normalizing uncertainty within LUP using the standard deviation across the entire dataset would be more effective. However, extracting features and updating for the entire dataset in each iteration is computationally unacceptable. Therefore, we use the standard deviation of each batch for normalization to optimize efficiency and accuracy.

**Qualitative ablation of LUP.** We further substantiate the rationale behind incorporating LUP through visual analysis. Fig. 4 presents the visualization of styles obtained using GSEE module and UGSDA method. It is observed that if only the GSEE module is used, the style distribution resembles that of the source domain. However, upon incorporating the LUP module, the distribution tends to disperse, encompassing a broader range of diverse styles. Given the invisibility of the target domains, this diversification is crucial for enhancing the network's generalization ability.

**Different data augmentation methods.** We also compare our method with various data augmentation methods. We first conduct data augmentation using the "random" setting from MixStyle [66]. Since MixStyle only considers source styles, its performance is inferior to ours. After that, we follow RobustNet [5] for photometric transformations. As a manually predefined data augmentation method, it struggles to encompass all possible styles. Hence, this approach underperforms on the UCF_QNRF dataset, which exhibits more significant domain shifts. Compared to these methods, our proposed method is validated to generate more diverse and realistic samples

**Table 6: Ablation studies of different data augmentation methods.**

| Method | SHB | | UCF_QNRF | |
|---|---|---|---|---|
| | MAE | MSE | MAE | MSE |
| MixStyle [66] | 12.6 | 25.7 | 127.5 | 221.2 |
| Photometric [5] | 13.5 | 25.0 | 132.6 | 226.2 |
| Ours | **11.6** | **24.5** | **117.0** | **194.1** |

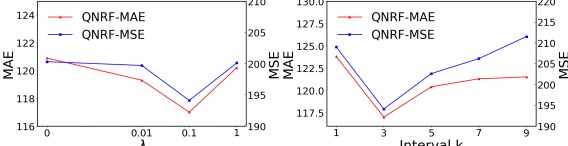

Figure 5: Quantitative evaluation with different $\lambda$ and interval $k$.

by comprehensively considering both the existing real styles in the source domain and potential unknown styles, thereby enhancing the model's generalization capability.

**Ablation of hyperparameters.** We determined the parameter $\lambda$ for the DDC loss through a series of ablation experiments. As depicted in the left graph of Fig. 5, a line chart illustrates the network performance variations with changes in $\lambda$, using SHA as the source domain and QNRF as the target domain. The optimal value of 0.1 was selected as the final parameter choice.

We then study the impact of the basis styles $\mu_{basis}$ and $\sigma_{basis}$ on network performance under varying update frequencies $k$. As shown in the right graph of Fig. 5, the model achieves the best performance with an update frequency $k = 3$. This is because when the update frequency is too rapid, the model becomes overly focused on the source domain, losing diversity. Conversely, a lower update frequency leads the model to deviate from the basis styles provided by the source domain, resulting in a loss of realism. Besides considering performance, the computational complexity is also feasible, making $k = 3$ as our chosen setting.

## 5 CONCLUSION AND FUTURE WORK

In this paper, we pioneeringly explore style-based data augmentation in the domain generalization of crowd counting. An uncertainty-guided style diversity augmentation method is proposed. To acquire diverse and real samples, the method employs a global style element extraction module to sample global style elements from the entire source domain, complemented by local uncertainty perturbations at the batch level for uncertainty styles estimation. Besides these, we introduce the density distribution consistency loss and blend to further constrain the density distributions of real and stylized samples, enhancing the realism of the stylized samples. Comprehensive experiments validate that our method effectively enhances the network's domain generalization capabilities across various datasets and network architectures.

In future work, we aim to investigate the domain generalization from synthetic to real datasets, a task that presents greater challenges due to the substantial domain gap and more pronounced style disparities between them.

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
