# OpenReview forum: "Domain-Agnostic Crowd Counting via Uncertainty-Guided Style Diversity Augmentation"
_acmmm.org/ACMMM/2024/Conference — MM2024 Poster_

### Official Review · Reviewer_jPm6 · 2024-05-11

**Rating:** 4
**Confidence:** 4

**Summary:**

The paper introduces an Uncertainty-Guided Style Diversity Augmentation (UGSDA) method to overcome the performance barriers posed by domain shift in crowd counting algorithms. As acquiring target domain images can be challenging, UGSDA enables crowd counting models to be trained exclusively on the source domain and generalize effectively to unseen target domains. This is achieved through the integration of three key components: Global Styling Elements Extraction, Local Uncertainty Perturbations, and Density Distribution Consistency loss. Extensive experiments validate the approach's strong generalization capabilities across various datasets and models.

**Strengths:**

1) The paper proposes a novel Uncertainty-Guided Style Diversity Augmentation (UGSDA) method that effectively addresses the domain shift challenge in crowd counting.
2) The integration of global styling elements extraction, local uncertainty perturbations, and density distribution consistency loss creates a robust framework that generates realistic and diverse samples during training.
3) The paper provides comprehensive experiments across various datasets and models, demonstrating the strong generalization capabilities of the UGSDA method. This adequate evaluation validates its effectiveness in addressing domain shift.

**Limitations:**

1) The paper mentions a complex computing evaluation compared to DGCC but does not elaborate on it. Clarification may be needed.
2) Fine-grained framework and visualizations analytics of the GSEE and LUP components are missing, which hampers understanding of the method.

**Suitability:**

2

---

### Official Review · Reviewer_m5cN · 2024-05-16

**Rating:** 4
**Confidence:** 3

**Summary:**

This paper proposes an Uncertainty-Guided Style Diversity Augmentation (UGSDA) method, enabling the crowd-counting models to be trained solely on the source domain and directly generalized to different unseen target domains. UGSDA generates diverse and realistic samples with a GSEE module, LUP module, and DDC loss where GSEE extracts global style elements, LUP obtains uncertainty perturbations, and DDC loss regulates the extent of perturbations.

**Strengths:**

- The first attempt is to employ data augmentation techniques for DG in crowd counting.
- Plug-and-play functionality without affecting inference speed.
- Extensive experiments with benchmarks on different scales demonstrate the effectiveness of the proposed method.

**Limitations:**

- **Insufficient Baselines**: It would be good to compare the proposed method with existing DG data augmentation methods such as CrossGrad, DDAIG, and MixStyle. Although these methods are always applied to image datasets, they can also be applied to different data types.

**Suitability:**

2

---

### Official Review · Reviewer_4FUF · 2024-05-23

**Rating:** 3
**Confidence:** 4

**Summary:**

The paper introduces an Uncertainty-Guided Style Diversity Augmentation (UGSDA) method to improve domain generalization in crowd counting by generating diverse training samples. UGSDA incorporates Global Styling Elements Extraction (GSEE), Local Uncertainty Perturbations (LUP), and Density Distribution Consistency (DDC) loss. While the approach shows promise, the paper lacks sufficient comparison with existing methods, clear visualization of style differences, and proof of its innovative aspects. Additionally, some figures are unclear.

**Strengths:**

1. The paper is well-structured, and the methods are described in a clear and understandable manner.
2. The paper includes a comprehensive set of experiments.
3. The experimental results are competitive and demonstrate the effectiveness of the proposed method.

**Limitations:**

1. Inference Speed Comparison: The inference speed of DGCC is not comparable to demonstrate advantages.
2. Style Unclear: Style is noticeable in many tasks but unclear in crowd counting. The paper does not visualize the style differences in different images or provide specific explanations of the styles.
3. Impact of Style Differences on Generalization: The paper needs to prove how style differences affect domain generalization performance.
4. Diversity of Source Domain Samples: The diversity of source domain samples limits the method's generalization ability.
5. Lack of Innovation: The method appears to be a transfer of techniques from other fields to the counting domain. The paper needs to compare this work with reference [63] to highlight its unique contributions.
6. Clarity of Figure 2: Figure 2 is unclear, and the FPS is not explained. Additionally, there is no visualization of style transformation before and after augmentation, making it hard to understand.

**Suitability:**

2

---

### Official Review · Reviewer_hu69 · 2024-05-31

**Rating:** 4
**Confidence:** 2

**Summary:**

The paper addresses the challenge of domain shift in crowd counting algorithms, which struggle to perform well in unseen domains. Traditional domain adaptation methods require images from the target domain and additional training time, making them impractical when target domain images are hard to acquire. The authors propose an Uncertainty-Guided Style Diversity Augmentation (UGSDA) method to overcome this limitation. UGSDA enables models to train solely on source domain images and generalize to unseen target domains by generating diverse and realistic samples during training.

**Strengths:**

This paper presents an approach in the field of crowd counting by employing data augmentation techniques for domain generalization. The method offers a simple, plug-and-play solution that does not impact inference speed. The proposed Uncertainty-Guided Style Diversity Augmentation (UGSDA) method dynamically generates diverse training samples by combining global style elements from the source domain with slight perturbations from each training batch. Furthermore, the Density Distribution Consistency (DDC) loss effectively ensures the realism of the density distribution in the stylized samples. This innovative approach enhances the model's generalization capabilities without the need for additional training time or target domain images.

**Limitations:**

In Table 1, the performance gain seems to be marginal on SHB with SHA as source domain.

**Suitability:**

2

---

### Meta-Review · Area_Chair_2yqX · 2024-07-03

**Recommendation:** Accept (Poster)
**Confidence:** 4

**Metareview:**

This submission presents the Uncertainty-Guided Style Diversity Augmentation (UGSDA) method to improve crowd counting algorithms' performance in unseen domains by generating diverse and realistic training samples solely from the source domain. The method, incorporating Global Styling Elements Extraction (GSEE), Local Uncertainty Perturbations (LUP), and Density Distribution Consistency (DDC) loss, demonstrates good generalization capabilities across various datasets and models.

All reviewers acknowledge that the proposed method is simple, clear, and achieves good performance. The initial concerns about unclear visualization, missing baseline, and clarity of component analysis were addressed in the rebuttal. Please ensure the response is included in the revised version, and make the code publicly available as promised.